# Nano-curcumin enhances the sensitivity of tamoxifen-resistant breast cancer cells via the *Cyclin D1-DILA1* axis and the *PI3K/AKT/mTOR* pathway downregulation

Taraneh Givi[1], Fatemeh Mohajerani[1], Zahra Moazezi Tehrankhah[1],
Mohammadjavad Karimi Taheri[1], Maryam Amirahmadi[2], Sadegh Babashah[1],
Majid Sadeghizadeh[1]*

1 Department of Genetics, Faculty of Biological Sciences, Tarbiat Modares University, Tehran, Iran,
2 Food and Drug Laboratory Research Center, Food and Drug Administration, MOH MOE, Tehran, Iran

* sadeghma@modares.ac.ir

## Abstract

This research investigates the effects of nano-curcumin on the expression of *Cyclin D1* and *DILA1* genes, as well as on the *PI3K/AKT/mTOR* pathway, in tamoxifen-sensitive and resistant MCF-7 breast cancer cell lines. The IC50 values were determined using the MTT assay. To establish tamoxifen-resistant MCF7 cells, a stepwise exposure to tamoxifen was conducted, gradually increasing the concentration to twice the IC50 dose. Following treatment with nano-curcumin, cell migration, proliferation, and apoptosis were evaluated using a wound healing assay and flow cytometry. Additionally, the impact of nano-curcumin on the *PI3K/AKT/mTOR and Cyclin D1* signaling pathways was analyzed via qRT-PCR and Western blot. Nano-curcumin treatment significantly inhibits cell proliferation, viability, and migration and promotes apoptosis in tamoxifen-sensitive and resistant MCF7 cell lines. qRT-PCR and Western blot analyses revealed reduced expression of *Cyclin D1, DILA1, NF-κB, PI3K, AKT, mTOR, VEGFα, MMP2,* and *BCL2*, along with increased levels of *PTEN, TIMP3, RECK,* and *BAX*. Our results indicate that nano-curcumin mitigates drug resistance by decreasing *DILA1* expression, which destabilizes Cyclin D1 protein and downregulates *PI3K/AKT/mTOR* pathway genes. Additionally, nano-curcumin induces apoptosis and inhibits migration. These findings suggest the nano-curcumin and tamoxifen combination may be a promising therapeutic strategy to overcome drug resistance in ER+ breast malignancies.

## Introduction

Worldwide, breast cancer affects one in every eight women, accounts for about 11.7 percent of all cancer cases, and is the second leading cause of cancer-related

**Data availability statement:** All data generated and analyzed during this study are original and are included in this article and its supplementary files.

**Funding:** This work is based upon research funded by Tarbiat Modares University Tarbiat Modares University as funders had no role in study design, data collection and analysis, decision to publish, or preparation of the manuscript, it provided the resource and equipment necessary for conducting the experiments.

**Competing interests:** No authors have competing interests.

**Abbreviations:** MCF7-S, MCF7 tamoxifen-sensitive; MCF7-TR, MCF7 tamoxifen resistance; MCF7-TR-NC, MCF7 tamoxifen resistance treated with nano-curcumin; WHO, World Health Organization; ANOVA, One-way analysis of variance; qRT-PCR, Quantitative real-time PCR; DILA1, Cyclin D1-Interacting Long noncoding RNA 1; CCND1, Cyclin D1; AKT, Protein kinase B; MTOR, MSammalian target of rapamycin; PI3K, Phosphoinositide 3-kinase; NF-kB, Nuclear factor kappa-B; TIMP3, Tissue inhibitor of metalloproteinase 3; RECK, Reversion Inducing Cysteine-Rich Protein with Kazal Motifs; MMP2, Matrix metalloproteinase-2; BAX, BCL2 associated X, apoptosis regulator; BCL2, BCL2 apoptosis regulator; CCNA2, Cyclin A2; CDKN1A, cyclin dependent kinase inhibitor 1A; PTEN, Phosphatase and Tensin Homolog; VEGF, A Vascular Endothelial Growth Factor A.

death in females [1]. The most common female cancer in Iran is breast cancer, which accounts for about 24.4% of all cases of cancer in women [2]. ER-positive (ER+) and ER-negative (ER-) breast cancer are two subtypes of breast cancer that can be distinguished based on whether the estrogen receptor (ER) is expressed or not. ER+ breast cancer is more prevalent and typically has a better prognosis, whereas ER- breast cancer is normally more aggressive and less responsive to hormone therapy [3].

Tamoxifen is a widely used selective estrogen receptor modulator (SERM) for the treatment of ER+ breast cancer. Administration of tamoxifen for ER+ treatment decreased the recurrence rate by an average of 39%, demonstrating the effectiveness of endocrine therapy in reducing the recurrence of early ER+ breast cancer. However, approximately one-third of these patients still experience de novo or acquired resistance, resulting in tumor relapse and metastasis, which makes resistance to tamoxifen a significant clinical challenge [4,5].

Mechanisms of tamoxifen resistance are complex and consist of alterations in ER signaling, activation of alternative growth factor pathways, and changes in cell cycle regulators [6]. Moreover, sustaining Proliferative Signaling, activating invasion and metastasis, and resisting cell death are three of the six hallmarks of cancer [7], which can relate to tamoxifen resistance. Identifying the resistance mechanisms and finding strategies to combat them is crucial to patients' survival.

The occurrence of tamoxifen resistance (TamR) in tamoxifen-sensitive (TamS) cells has been extensively studied using cultured cancer cells or continuous treatment of cancer cells with tamoxifen. The widely used MCF7 breast cancer cell line is a prominent model for this research. Cancer cells exhibit the activation of alternative pathways to develop TamR and sustain proliferation in the face of ER pathway inhibition. Notably, the *PI3K/Akt/mTOR* pathway, a downstream signaling cascade, can be triggered by various stimuli in the absence of estrogen signaling, facilitating cell growth and survival even in the presence of tamoxifen. Among long non-coding RNAs (lncRNAs), several genes have been identified as contributors to TamR, each operating via unique mechanisms. Noteworthy among these is HOTAIR, which has shown the capability to stimulate the *PI3K/Akt/mTOR* signaling pathway, thereby promoting TamR [8]. *DILA1* is a new lncRNA that interacts with *Cyclin D1* and is overexpressed in tamoxifen-resistant breast cancer cells [5].

Several mechanisms of tamoxifen resistance have been investigated, including the overexpression of *Cyclin D1*, a cell cycle regulator that promotes cell proliferation, which is overexpressed in a subset of breast cancers [9]. The *Cyclin D1* gene plays a crucial role in cell cycle regulation by G1-S transition induction through Cdk4 and Cdk6 activation and has a role in the development and progression of breast cancer. Overexpression of *Cyclin D1* is observed in around 50% of breast cancer cases, while the *CCND1* amplification frequency is between 9–15%, and both are associated with a poor prognosis [10]. Moreover, *Cyclin-D1* overexpression has been linked to tamoxifen resistance in ER-positive breast cancer, suggesting that it may be a potential therapeutic target [11].

Blocking *cyclin D1* can hinder estrogen-driven cell growth, while increased *cyclin D1* levels can counteract the effects of antiestrogen-induced growth

arrest. Additionally, growth factors that stimulate *cyclin D1* production may facilitate tamoxifen resistance. In *HER2*-overexpressing tamoxifen-resistant breast cancer cells, tamoxifen-induced growth was linked to activation of the *PI3K/Akt* and *MAPK/ERK1,2* pathways and *cyclin D1* expression [12]. Targeting these pathways and genes may provide new therapeutic options for breast cancer patients [13]. The crucial role of the *PI3K/Akt/mTOR* pathway is proven in cell growth and survival, glucose metabolism, and protein synthesis [14]. Furthermore, dysfunction of this pathway is observed in various types of tumors, including triple-negative breast cancer, leading to research on inhibiting this pathway in cancers. It is believed that blocking the *PI3K/AKT/mTOR* pathway could enhance ER activity, contributing to resistance [15,16].

Curcumin, a potent natural anticancer agent, demonstrates a range of pharmacological activities in clinical research. However, its therapeutic efficacy is hindered by limitations such as poor solubility, rapid degradation, and low absorption rates of its hydrophobic molecules in vivo. Recent advancements have explored various nanocarriers to enhance the bioavailability of this hydrophobic biomaterial, thereby improving its clinical applicability [17,18]. The utilization of nano-formulation herbal medicines for cancer therapy represents a promising area of research with significant potential [19,20]. Nano-curcumin, a nanoparticle formulation of curcumin, has gained attention for its potential anti-cancer properties. Studies have shown that Dendrosomal curcumin (nano-curcumin) exhibits enhanced bioavailability and improved therapeutic efficacy compared to free curcumin, and it has powerful anti-cancer potential [21]. The dendrosomal nano-curcumin (DNC) formulation used in this study was prepared as previously described by Tahmasebi et al. The nanoparticles had an average particle size of approximately 142 nm, a polydispersity index (PDI) of 0.22, and a zeta potential of –7.4 mV, indicating moderate colloidal stability and uniform dispersion. The same batch of DNC was used throughout all experiments to ensure consistency and reproducibility, despite the fact that nano-curcumin used in this study was stable during the experiment, determination of batch stability was performed every 6 months [22].

Recent research has demonstrated that nano-curcumin can inhibit the growth of breast cancer cells, induce apoptosis, and modulate signaling pathways involved in tamoxifen resistance, suggesting its potential as a novel therapeutic agent for breast cancer treatment [23]. In a recent study, PEGylated curcumin derivative attenuated breast cancer progression by regulating *Cyclin-D1* and *p53* [24].

The primary objective of our study was to elucidate the effects of nano-curcumin on continuously treated cancer cells with tamoxifen (MCF7 Tamoxifen Resistant cell line) and investigate the underlying curcumin mechanisms involving *Cyclin D1, DILA1*, and the *PI3K/AKT/mTOR* signaling pathway in both MCF7 Sensitive (MCF7-S) and MCF7 Tamoxifen Resistant (MCF7-TR) cell lines. Our findings demonstrated that nano-curcumin significantly reduced cell proliferation and migration in both cell line types. Additionally, nano-curcumin contributed to the activation of apoptotic pathways and induction of cell cycle arrest in the sub-G1 phase, even in resistant cells.

## Materials and methods

### Cell lines and culture conditions

MCF7 (Human breast cancer cell line) and MCF10A (Normal human mammary epithelial cell line), procured from the Cell Bank of Iran at the Pasteur Institute in Tehran, underwent authentication through short tandem repeat amplification, and the cells tested negative for mycoplasma contamination.

This study was approved by the Ethics Committee of Tarbiat Modares University (No. IR.MODARES.REC.1401.073). MCF7 and MCF10A cells were cultured in Dulbecco's Modified Eagle's Medium (DMEM) (Gibco, USA) medium, supplemented with 10% fetal bovine serum (FBS; Hyclone, USA). These cells were incubated in a humidified atmosphere at 37˚C with 5% $CO_2$. Sub-culturing was initiated upon reaching 80% confluence. MCF7-TR cells were derived from MCF7 cells by continuous exposure to TAM diluted in DMSO. Increasing concentrations of TAM were consistently administered to MCF7 cells for six months, as described in S1 Table. The initial treatment involved a 0.5µM TAM concentration, followed by an incremental increase of 0.25µM per week.

## Cell viability assay (MTT assay)

For the MTT assay, $5 \times 10^3$ MCF7-S, MCF7-TR, and MCF10A cells were seeded in 96-well plates in triplicate. After 24h, the MCF7-S cells were treated with TAM concentration gradient, and the MCF7-TR and MCF10A cells were treated with nano-curcumin (MCF7-TR-NC, MCF10A-NC). Although the same batch of nano-curcumin was used in all experiments, no batch variation occurred, and the formulation employed was derived from a single, well-characterized stock solution to ensure consistency and reproducibility of the results. Nano-curcumin is a polydisperse colloidal suspension formulated in dendrosomes. This formulation enables it to be directly dispersed in aqueous solutions or cell culture media without requiring organic solvents such as DMSO. Therefore, in all nano-curcumin–related experiments, no additional vehicle is needed to be used.

At 24h and 48h post-treating, 10 μL of 5 mg/mL in PBS Tetrazolium salt (Sigma-Aldrich, USA) was added to each well, followed by incubation at 37°C for 4 hours. The supernatants were discarded once the purple precipitate appeared in the wells. Subsequently, 100 μL of DMSO (Merk, USA) was added to each well to dissolve the formazan crystals, and the optical density (OD) was measured at 570 nm using an ELISA Microplate Reader (BioTek, USA).

## RNA isolation, cDNA synthesis, and real-time PCR (qRT-PCR)

Total RNAs were extracted from the cell lines with Trizol Reagent (Invitrogen, USA). RNA quality was assessed through 1% agarose gel electrophoresis, showing the 28S and 18S ribosomal.

RNA (rRNA). The RNA concentration was evaluated by a Nanodrop ND-1000 spectrophotometer (Thermo Fisher Scientific, USA), and the sample's 260/280 OD ratio was $2.0091 \pm 0.001$ (Mean±SEM), indicating their high purity. After treating 1 μg of each RNA sample with DNase I RNase-free (Thermo Scientific, USA) at 37°C for 30 min to remove any contaminating genomic DNA, cDNA synthesis followed a standard protocol using a cDNA synthesis kit (Biotechrabbit, Germany) in a 20 μl reaction, and stored at −20°C until use.

RT-PCR was conducted using 1 μL of cDNA template along with Phusion High-Fidelity PCR Master Mix (Thermo Fisher Scientific, USA) and 0.4 μM of each primer in a 20 μL PCR reaction following the manufacturer's instructions. Real-time PCR was performed using SYBR® Premix Ex TaqTM II (TAKARA, Japan) in the ABI Step One Sequence Detection Real-time PCR system (Applied Bio-systems, CA, USA). The internal control used was *GAPDH*.

Quantitative measurement of *CyclinD1, DILA1, BAX, BCL2, AKT, PTEN, PI3k, P53, TWIST, E-cad, c-Myc,* and *mTOR* gene expression was performed. The primer sequences for the genes analyzed were designed with the Eurofins Genomics Primer Design Tool (https://eurofinsgenomics.eu) and validated with an IDT oligo analyzer (https://www.idtdna.com) and NCBI Primer blast (https://www.ncbi.nlm.nih.gov). Specific primers utilized in the study can be found in S2 Table. The relative expression levels and fold changes were determined using the $2^{-\Delta Ct}$ and $2^{-\Delta\Delta Ct}$ formulas, respectively. GraphPad Prism software version 9.3 was utilized to assess the significance of the findings.

## Cell cycle analysis

Cell cycle analysis was performed using the flow cytometry technique to confirm TAM resistance. MCF7-S and MCF7-TR cells ($2 \times 10^3$ cells/well) were cultured in triplicates in 6-well plates. After 24h, a group of wells were treated with 6 μM tamoxifen, while other groups were treated with 26 μM nano-curcumin. At 48h post-treatment, the cells were harvested, centrifuged at 1300 rpm for 5 min, washed twice with PBS, and fixed in 70% cold ethanol at 4°C for 2 hours. The fixed cells were then stained with a PBS solution containing 50 μg/mL propidium iodide (Sigma, USA), 100 μg/mL RNase A (BioBasic, Canada), and 0.1% Triton X-100 (Sigma, USA), followed by a 30-min incubation at room temperature in the dark. Subsequently, all samples' cellular DNA content was analyzed using a FACS Calibur Flow Cytometer (BD biosciences, USA), and FlowJo software was utilized for data interpretation.

## Apoptosis analysis

Cell apoptosis was evaluated utilizing the Annexin V-PI staining kit following the manufacturer's guidelines (Roche, Germany). MCF7-S, MCF7-TR, and MCF10A cells were seeded ($2 \times 10^3$ cells/well) in 6-well plates in triplicate and treated with NC as previously described. At 48h after NC treatment, cells were rinsed with cold PBS, resuspended in 1X binding buffer (10 mM HEPES/NaOH, pH 7.4, 140 mM NaCl, 2.5 mM CaCl2), and then stained with Annexin V-PI in the absence of light for 15 min at room temperature. Apoptosis and necrosis in the cells were assessed using the FACS Calibur Flow Cytometer (BD Biosciences, USA), and the data were analyzed using FlowJo software.

## Protein extraction and Western blotting

MCF7-S, MCF7-TR, and MCF7-TR-NC cells were lysed using RIPA buffer (BioBasic, Canada) on ice for 30 min as per the manufacturer's instructions. Total protein concentration was measured using the Bradford assay. The Bradford reagent was prepared by dissolving Coomassie blue G-250 (Merck, Germany) in methanol (Merck, Germany) and 85% phosphoric acid (H3PO4) (Merck, Germany). This solution was then added slowly to water and filtered using Whatman paper No1 (Merck, Germany). The Bradford reagent was added to each sample to assess the concentration of standard protein (BSA) and protein samples, and absorbance was measured at 630 nm using a spectrophotometer. Total protein was separated by SDS-PAGE on a 10% polyacrylamide gel and transferred to a PVDF membrane (Thermo Fisher Scientific, USA). The membrane was blocked at 4°C for 2 hours using 5% skim milk (Sigma, USA) diluted in PBST (BioBasic, Canada). Subsequently, the membrane was incubated overnight at 4°C with primary antibodies against cyclin D1 molecular weight: 37 kDa (dilution ratio 1: Santa Cruz Biotechnology Cat# sc-8396), Akt molecular weight: 62 kDa (dilution ratio 1: Santa Cruz Biotechnology Cat# sc-5298), BAX molecular weight: 23 kDa (dilution ratio 1: Santa Cruz Biotechnology Cat# sc-7480),BCL2 molecular weight: 26 kDa (dilution ratio 1: Santa Cruz Biotechnology Cat# sc-7382),PI3K molecular weight: 85 kDa (dilution ratio 1: Santa Cruz Biotechnology Cat# sc-515646),mTOR molecular weight: 289 kDa (dilution ratio 1: Santa Cruz Biotechnology Cat# sc-517464)and p-mTOR molecular weight: 289 kDa (dilution ratio 1: Santa Cruz Biotechnology Cat# sc-293089) and p-Akt molecular weight: 60 kDa, at specified dilutions. Following primary antibody incubation, the membrane was probed with mouse anti-rabbit IgG-HRP (sc-2357) or m-IgGκ BP-HRP (sc-516102) secondary antibodies as per the manufacturer's instructions and incubated for 1 hour after washing with PBST. Protein signals were visualized using an ECL detection reagent (Thermo Fisher Scientific, USA), with GAPDH protein as the normalization control. The orginal Western blot images are available in S1 Fig.

## Wound healing assay

To assess the migration inhibitory effects of the nano-curcumin, MCF7-S and MCF7-TR cells were seeded ($5 \times 10^4$ cells/well) in a 24-well plate in triplicates. After 24h of incubation, the cells were treated with NC. At 24h post-treatment, a straight linear wound was made across the confluent monolayer cell in each well using a sterile 100 μl pipette tip. In this step, the plucked cells were washed out with PBS. In the subsequent stage, the cells were cultured in DMEM with 10% serum (FBS) at 37°C with 5% CO2. The cell migration areas were scanned by measuring the distance between the two sides of the scratch after 24h and 48h using an inverted microscope (OLYMPUS IX53), and the images were analyzed with ImageJ software.

## Statistical analyses

All experiments were conducted independently in triplicate, and results are presented as means ± standard error of the mean (SEM). P-values less than 0.05 were deemed statistically significant. Statistical analyses were performed using GraphPad Prism 9.4 software, employing the t-test for comparisons between two groups and One-Way Analysis of Variance (ANOVA) for comparisons involving three or more groups to assess significant differences in the measured variables.

## Results

### Generation of MCF7 tamoxifen-resistant cells

To generate tamoxifen-resistant MCF7 cells, we used tamoxifen citrate [25]. The drug sheet suggested three solvents for tamoxifen: Ethanol, Water, and DMSO. Considering better solubility and lower cytotoxicity, we chose the DMSO solvent. DMSO's potential cytotoxicity was evaluated to ensure that the used concentration does not have any impact on cell viability. MTT assay at 24h and 48h, also Annexin V/PI staining after 48h, confirmed that DMSO at a final concentration of 0.016 μM had no cytotoxic effects on cells, ensuring that any observed cytotoxicity was solely attributed to tamoxifen (Fig 1A, B). Therefore, the stock solution was prepared at a concentration of 5 mM.

To determine the $IC_{50}$, we conducted an MTT assay at 24h and 48h. $IC_{50}$ value for Tamoxifen in MCF7-S cells was 6.42 μM at 24h and 6.1 μM at 48h. (Fig 1C, D). Based on S1 Table, we started with a concentration of 0.5 μM to induce resistance and continued this process up to 2-fold the $IC_{50}$ dose for six months (Tamoxifen has been constantly present in the cell culture). After 6 months, their morphology was examined using a light microscope to confirm that the MCF7 cells had developed tamoxifen resistance (Fig 1E). Subsequently, the expression levels of *PI3K, AKT1, mTOR, Cyclin D1, DILA1, P53, c-Myc*, and *EMT* genes were assessed by qRT-PCR. Additionally, the apoptosis and cell cycle were measured by flow cytometry, and the protein levels of AKT, p-AKT, and Cyclin D1 were determined using Western blot.

The qRT-PCR results showed that, compared to MCF7-S cells, the expression levels of *PI3K, mTOR, c-Myc, Cyclin D1,* and *DILA1* statistically increased. In contrast, the *E-Cadherin, Twist,* and *P53* mRNA expression levels are down-regulated in MCF7-TR (Fig 1F). Furthermore, the cell cycle analysis after 48h demonstrated a significantly higher percentage of S and G2/M cells in MCF7-TR compared to MCF7-S (Fig 1G, H). According to the apoptosis analysis of the cells treated with the tamoxifen citrate for 48h, the apoptosis rate notably increased in MCF7-S cells compared to MCF7-TR cells (Fig 1I, J). AKT, p-AKT, and Cyclin D1 protein levels were examined to further support that the cells had become resistant to tamoxifen. The results indicated an increased expression of these three proteins (1.13, 1.06, and 1.32-fold increase, respectively) in MCF7-TR compared to MCF7-S (Fig 1K). Overall, these results indicate that MCF7 cells had become resistant to tamoxifen.

### Selecting the effective dosage of nano-curcumin on MCF7-S, MCF7-TR, and MCF10A cells

The MTT assay was used to determine the effective dosage of nano-curcumin on three cell lines, MCF7-S, MCF7-TR, and MCF10A, at 24h and 48h after treatment. The cell survival rate relative to the control group was examined using nano-curcumin at 10–60 μM concentrations. As shown in Fig 2A and B, the $IC_{50}$ value for dendrosomal nano-curcumin in MCF7-S cells was 23.34 μM at 24h and 22.96 μM at 48h. Additionally, the $IC_{50}$ value for dendrosomal nano-curcumin in MCF7-TR cells was 32.94 μM after 24h and 30.73 μM after 48h (Fig 2C, D). Furthermore, the cell viability rate of MCF10A cells was evaluated at 24 and 48 hours after treatment with nano-curcumin. These findings suggest that nano-curcumin does not induce significant cytotoxicity in MCF10A cells at concentrations lower than 40 μM, as illustrated in Fig 2E and F.

### Differential expression of *DILA1* and *Cyclin D1* genes in MCF7-S, MCF7-TR, and MCF10A cell lines

The expression levels of *DILA1* and *Cyclin D1* genes were assessed in MCF7-S, MCF7-TR, and MCF10A cells using qRT-PCR. The results corroborated that the expression level of the *DILA1* gene was significantly increased in MCF7-S and MCF7-TR cells compared to the MCF10A cell line ($p < 0.0001$)(Fig 3A). The *Cyclin D1* gene expression was also significantly up-regulated in MCF7-S and MCF7-TR cell lines compared to MCF10A cells ($p < 0.0004$)(Fig 3B). These results indicate that expression levels of DILA1 and Cyclin D1 genes are highest in tamoxifen resistant cells.

### Nano-curcumin decreased the cell viability and proliferation in MCF7-S and MCF7-TR cells compared to the MCF10A cell line

To investigate the impact of nano-curcumin on breast cancer cell lines' survival, an MTT assay was conducted at 24h, 48h, 72h, and 96h. The MCF7-S and MCF7-TR light and fluorescent microscope images before and after nano-curcumin

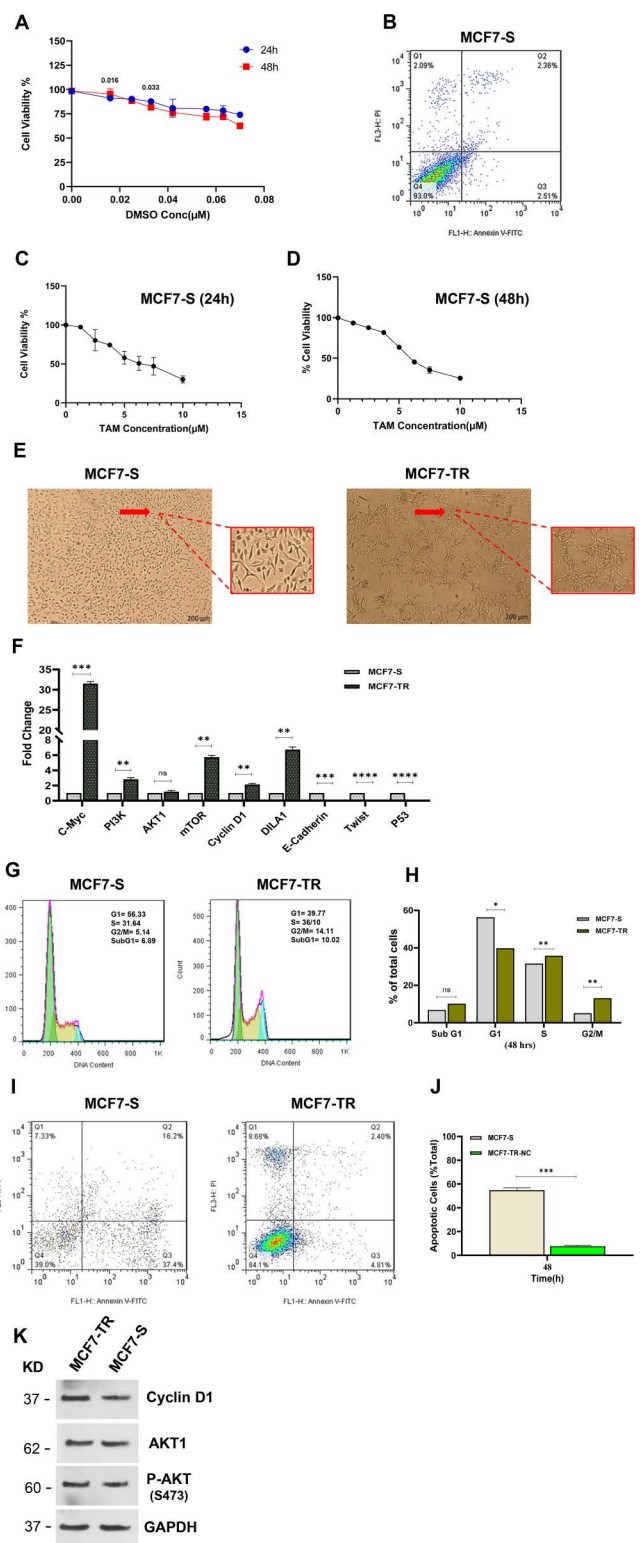

**Fig 1. Generation of MCF7-TR cells.** (A) Cell viability by MTT assay. The MTT assay confirmed that DMSO had no cytotoxic effects. (B) The apoptosis analysis of the MCF7-S after 48h confirmed that DMSO at a final concentration of 0.016 µM had no cytotoxic effects on cells (C, D) Cell viability by MTT assay. An $IC_{50}$ value for MCF7 cells after 24h and 48h. (E) The MCF7-S and MCF7-TR light microscope images. (F) qRT-PCR results of the *PI3K*, *AKT*, *mTOR*, *c-Myc*, *Cyclin D1*, *DILA1*, *E-Cadherin*, *Twist*, and *P53* genes mean fold change in MCF7-S compared to MCF7-TR. (G, H) The analysis of

the MCF7-TR cell cycle 48h after exposure to 6 μM of tamoxifen compared to MCF7-S, can be illustrated using both a histogram and bar plot. (I, J) The analysis of the MCF7-TR apoptosis rate 48h after exposure to 6 μM of tamoxifen compared to MCF7-S can be illustrated using both a histogram and bar plot. (k) Western blot analysis of AKT, p-AKT, and Cyclin D1 protein levels in MCF7-TR cells and MCF7-S. GAPDH was used as an endogenous control. The protein levels of AKT, p-AKT, and Cyclin D1 were increased in MCF7-TR cells compared to untreated cells. Columns and points, mean of three different experiments. Statistical analysis was conducted using a student t-test or one-way ANOVA, and means±SEM were displayed. ns = not significant, *$p < 0.05$, **$p < 0.01$, ***$p < 0.001$, ****$p < 0.0001$.

treatment (Fig 4A, B). The findings revealed that the viability of MCF7-S and MCF7-TR cells significantly decreased after nano-curcumin treatment compared to untreated cells at all time points evaluated ($p < 0.0001$) (Fig 4C, D).

The qRT-PCR analysis of *PI3K*, *AKT1*, *mTOR*, *NF-κB, Cyclin D1, DILA1,* and *PTEN* gene expression in MCF7-S and MCF7-TR cells showed that nano-curcumin was linked with a significant reduction in the mRNA levels of *PI3K*, *AKT1, mTOR, Cyclin D1, DILA1,* and *NF-κB* ($p < 0.0001$) (Fig 4E, F). Additionally, the expression levels of *PI3K, AKT1, mTOR, NF-κB,* and *Cyclin D1* genes in MCF10A cells were assessed using qRT-PCR, both prior to and following treatment with nano-curcumin. The results indicated that no significant alterations in the expression of these genes before and after nano-curcumin treatment (Fig 4G).

The expression of the *PTEN* gene was significantly upregulated after nano-curcumin treatment in both MCF7-S and MCF7-TR cell lines ($p < 0.0001$) (Fig 4H, I). To further support the idea that nano-curcumin can reduce proliferation, the protein levels of mTOR, p-mTOR, PI3K, and Cyclin D1 were assessed using Western blot analysis in the MCF7-TR cell line after nano-curcumin treatment. The results demonstrated that the protein levels of p-mTOR, Cyclin D1, and PI3K were reduced in the MCF7-TR cell line treated with nano-curcumin compared to untreated cells (showing reductions of 0.55, 0.39, and 0.70-fold, respectively) (Fig 4J). Collectively, these results confirm that nano-curcumin is associated with decreased cell viability and proliferation by inhibiting *Cyclin D1*, *DILA1,* and the *PI3K/AKT1/mTOR* pathway.

### Nano-curcumin arrested the cell cycle and promoted apoptosis in MCF7-S and MCF7-TR cell lines through the suppression of *Cyclin D1* and *DILA1* gene expression

In the present study, we evaluated the potential impact of nano-curcumin on the cell cycle and apoptosis of breast cancer cell lines. The inhibitory role of nano-curcumin on the MCF7-S and MCF7-TR cell cycle was investigated by cell cycle analysis at 48h after treatment. Further, the *CCND1* (*Cyclin D1*), *CCNA2* (*Cyclin A2*), *DILA1*, and *CDKN1A* (*P21*) cell cycle gene expression levels were measured in the MCF7-S and MCF7-TR cell lines. Cell cycle analysis demonstrated that nano-curcumin treated cells compared to untreated cells is associated with the arrest of the cell cycle in the sub-G1 phase, with a concomitant decline in the proportion of S phase at 48h in MCF7 Sensitive treated with Nano-Curcumin (MCF7-S-NC) ($p < 0.0138$) (Fig 5A, B) and MCF7 Tamoxifen Resistant treated with Nano-Curcumin (MCF7-TR-NC) cells ($p < 0.0037$) (Fig 5C, D). The qRT-PCR results showed that nano-curcumin statistically significantly decreased the mRNA expression levels of *CCND1*, *CCNA2,* and *DILA1* in MCF7-S-NC ($p < 0.0001$) and MCF7-TR-NC cell lines ($p < 0.0001$). Evaluation of the expression level of *CDKN1A* in MCF7-S and MCF7-TR cell lines after being treated with nano-curcumin revealed that the *CDKN1A* gene expression level is statistically significantly increased in both cell lines compared to the untreated cells ($p < 0.001$) (Fig 5E, F).

Considering a notable rise in the sub-G1 phase, we utilized the Annexin-PI technique to examine the apoptosis rate in MCF7-S, MCF7-TR, and MCF10A cell lines at 48h after being treated with nano-curcumin. Additionally, the expression levels of apoptotic and anti-apoptotic genes and proteins were assessed using a qRT-PCR and western blot analysis in breast cancer cell lines treated with nano-curcumin. Fig 5G–J demonstrate that the percentage of MCF7-S and MCF7-TR apoptotic cells treated with nano-curcumin was significantly increased compared to untreated cells. Moreover, cell apoptosis analysis demonstrated that the level of cell death in MCF10A cells following 48 hours of nano-curcumin treatment remains negligible compared to the MCF10A-untreated cell line (Fig 5k, L).

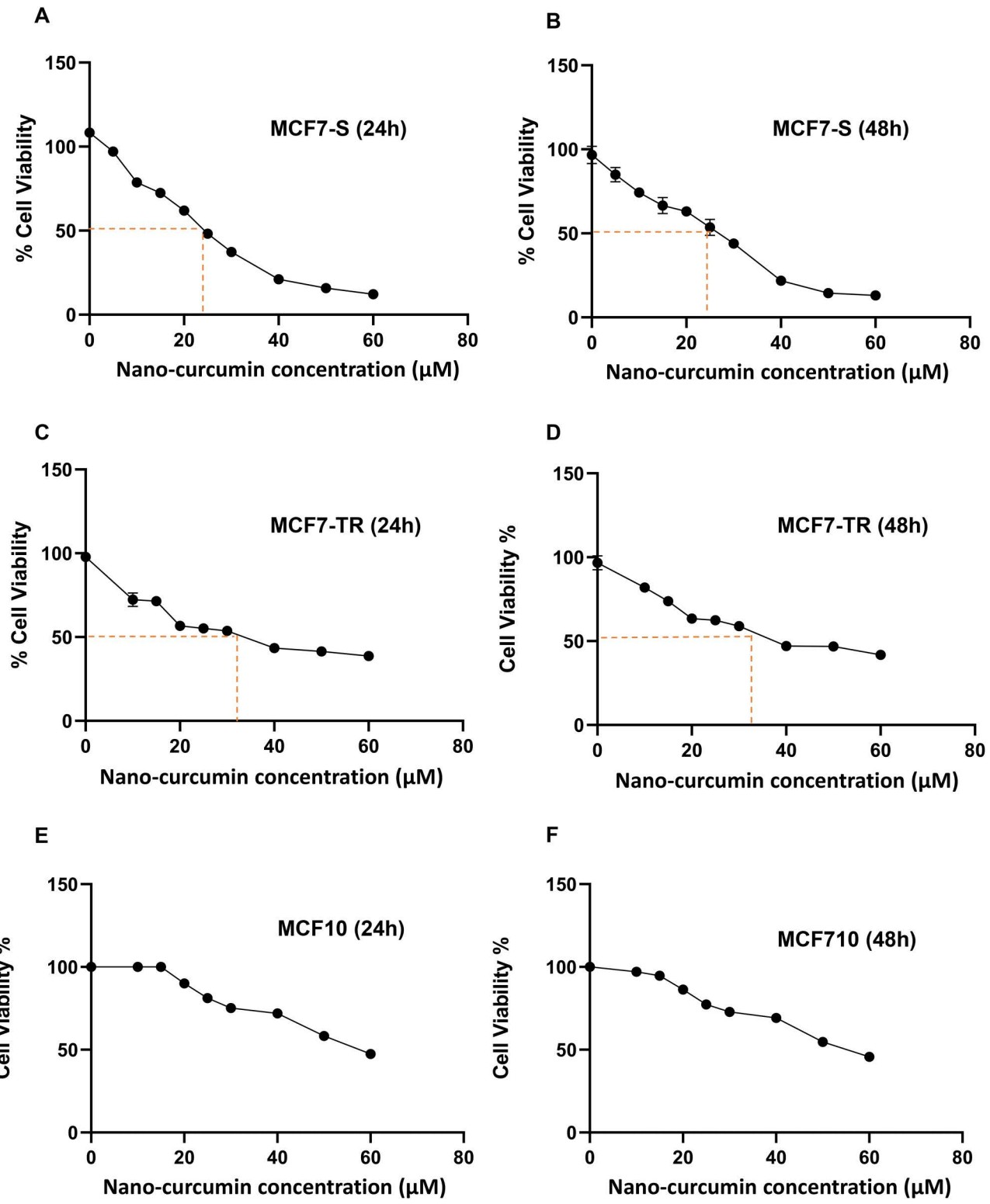

**Fig 2. Selecting the effective dosage of nano-curcumin.** (A, B) Cell viability by MTT assay. An $IC_{50}$ value for MCF7-S cells after 24h and 48h. (C, D) An $IC_{50}$ value for MCF7-TR cells after 24h and 48h. (E, F) Cell viability rate at 24h and 48h after nano-curcumin treatment on MCF10A cells. Points, mean of three different experiments. Statistical analysis was conducted using a student t-test or one-way ANOVA, and means±SEM were displayed. ns=not significant, *$p < 0.05$, **$p < 0.01$, ***$p < 0.001$, ****$p < 0.0001$.

 

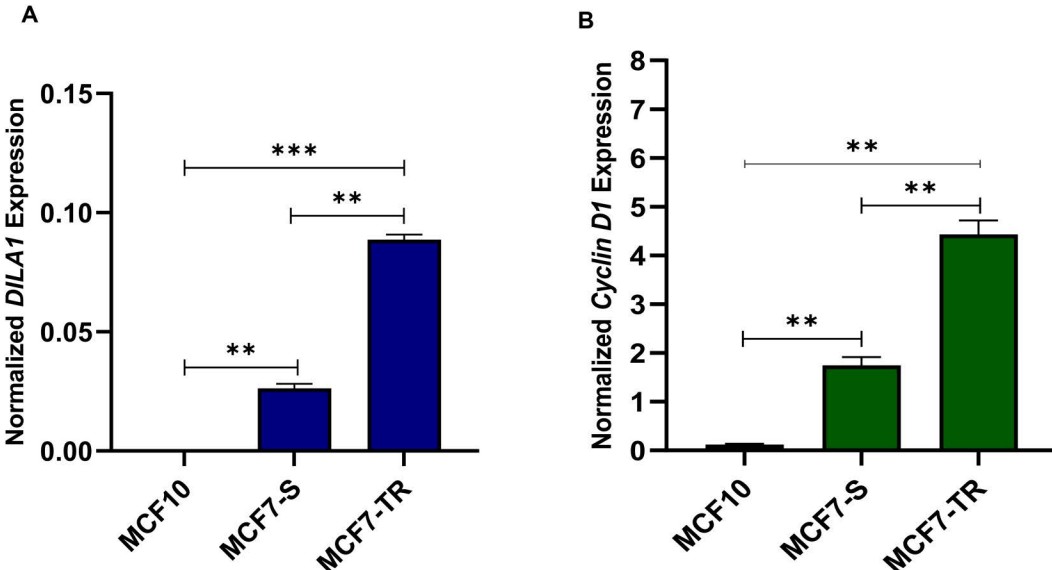

**Fig 3. The expression levels of *DILA1* and *Cyclin D1* genes.** (A) The *DILA1* gene expression was significantly increased in MCF7-S and MCF7-TR cells compared to the MCF10 cell line. (B) The expression level of the *Cyclin D1* gene was significantly up-regulated in MCF7-S and MCF7-TR cell lines compared to MCF10A cells. Columns represent the mean of three independent experiments. Statistical analysis was performed using a one-way ANOVA or student t-test, with means ± SEM presented. ns = not significant, *p < 0.05, **p < 0.01, ***p < 0.001, ****p < 0.0001.

Furthermore, nano-curcumin notably inhibited the mRNA levels of anti-apoptotic *BCL2* (*p < 0.004*) while statistically considerably up-regulating the level of *BAX* and *P53* mRNA (*p < 0.0001*) (Fig 5M, N). In addition, the protein level of BCL2 significantly decreased and the BAX protein level notably increased. Cells treated with nano-curcumin exhibited a marked elevation in the BAX/BCL2 ratio (11.41-fold) compared to the untreated cells (Fig 5O). These findings indicate that nano-curcumin can arrest cells in the sub-G1 phase of the cell cycle and induce apoptosis in breast cancer cell lines by inhibiting *Cyclin D1* and *DILA1* genes.

### Nano-curcumin inhibited cell migration in MCF7-S and MCF7-TR cells

To assess the effect of nano-curcumin on breast cancer cell migration in vitro, we investigated its effect on the migration of TamR cells using a wound healing assay and qRT-PCR analysis. Firstly, the cells' movement was monitored at different time points (0, 24, and 48h) in MCF7-S and MCF7-TR untreated and treated with nano-curcumin cell lines. Results showed that nano-curcumin decreased cell migration in MCF7-S and MCF7-TR cell lines treated by nano-curcumin toward the scratch wound compared to untreated cell lines (MCF7-S and MCF7-TR) (Fig 6A, C) and statistically significantly reduced the number of migrative cells in both cell lines after 48h (*p < 0.001*) (Fig 6B, D).

Secondly, to determine the nano-curcumin effect on cell migration, the expression level of *VEGFα, MMP2, TIMP3,* and *RECK* genes was evaluated in MCF7-S, MCF7-S-NC, MCF7-TR, and MCF7-TR-NC cells through amplification of related cDNA samples using qRT-PCR. Nano-curcumin in both MCF7-S-NC and MCF7-TR-NC cells led to a significantly decreased expression of the *VEGFα* and *MMP2* genes (p < 0.0001) and an increased expression of the *RECK* and *TIMP3* genes (*p < 0.0003*) (Fig 6E, F). These results confirmed that nano-curcumin significantly reduces the migration potential of MCF7-S and MCF7-TR breast cancer cells.

## Discussion

Breast cancer is the first-ranked contributor to cancer-related women's mortality and one of the most commonly diagnosed cancers worldwide. Estrogen receptors are the transcription factors responsible for approximately 75% of breast cancers,

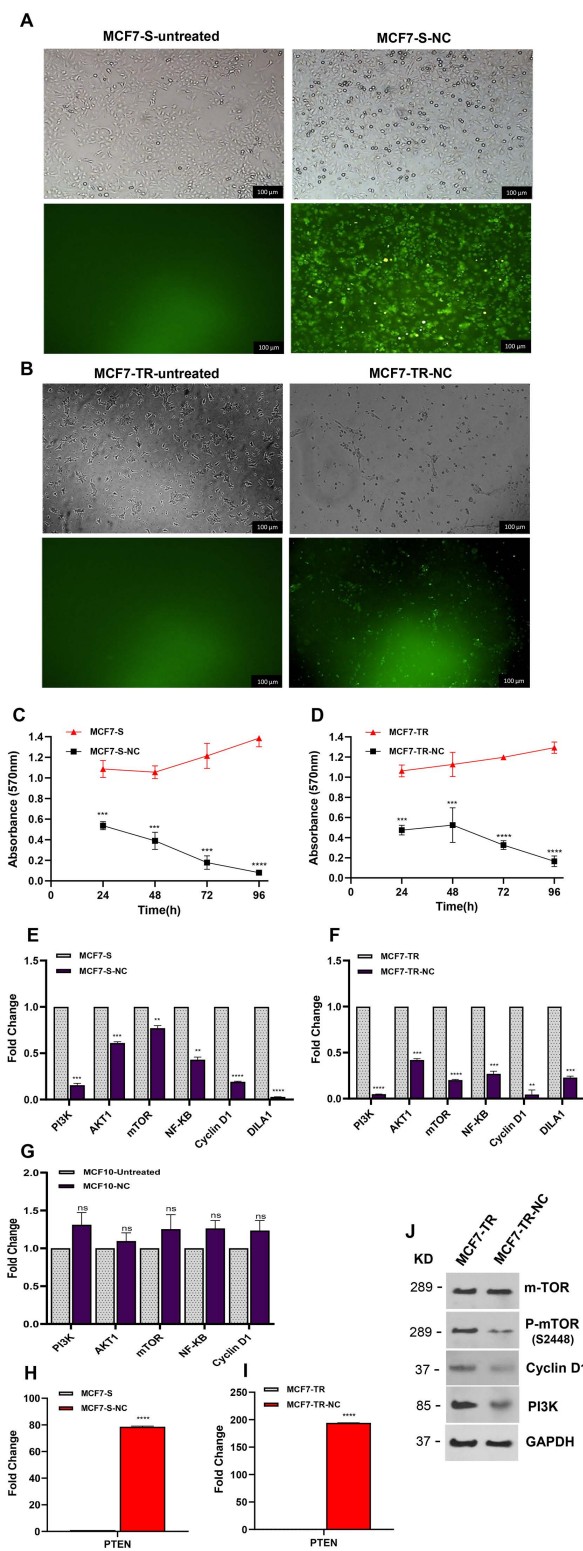

**Fig 4. The impact of nano-curcumin on cell viability and proliferation in breast cancer cells.** (A, B) The MCF7-S and MCF7-TR light and fluorescent microscope images before and after nano-curcumin treatment. (C, D) MTT assays were conducted on MCF7-S and MCF7-TR cells following nano-curcumin treatment. Cell viability significantly decreased at all tested time points. (E, F) mRNA Expression levels *PI3K*, *AKT1*, *mTOR*, *NF-KB*,

*Cyclin D1*, *DILA1* and *PTEN* in breast cancer cells. Nano-curcumin treatment led to a reduction in *PI3K*, *AKT1*, *mTOR*, *NF-KB*, *Cyclin D1*, and *DILA1* mRNA levels. (G) mRNA Expression levels *PI3K*, *AKT1*, *mTOR*, *NF-KB*, and *Cyclin D1* in MCF10. No significant alterations were observed in the expression of *PI3K*, *AKT1*, *mTOR*, *NF-KB*, and *Cyclin D1* genes before and after nano-curcumin treatment. (H, I) qRT-PCR analysis revealed up-regulation of the *PTEN* mRNA level in MCF7-S and MCF7-TR cells after nano-curcumin treatment. (J) Western blot analysis of mTOR, p-mTOR, and PI3K protein levels in MCF7-TR cells treated with nano-curcumin. GAPDH was used as an endogenous control. The protein levels of p-mTOR, Cyclin D1, and PI3K were reduced in MCF7-TR cells following nano-curcumin treatment compared to untreated cells. Columns and points represent the mean of three independent experiments. Statistical analysis was performed using a one-way ANOVA or student t-test, with means ± SEM presented. ns = not significant, *p < 0.05, **p < 0.01, ***p < 0.001, ****p < 0.0001.

making them an important goal for targeted endocrine therapies. Regardless of tamoxifen's effectiveness on ER-positive breast cancer patients as a notable antiestrogenic medication, the emergence of tamoxifen resistance represents a significant clinical challenge that restricts the success of endocrine therapies [26,27]. Therefore, finding novel methods to reverse tamoxifen resistance in breast cancer cells leads to making them impressible to therapeutic interventions, potentially enhancing the efficacy of treatments for resistant breast cancer.

The investigated resistance mechanisms emphasize the complex nature of estrogen receptor (ER) signaling and its interaction with crucial signaling pathways within breast cancer cells. Endocrine therapies have displayed efficacy by targeting key elements of cell signaling and cell cycle pathways, such as the *PI3K/AKT/mTOR* and the Cyclin D/cyclin-dependent kinase 4 and 6 pathways [16].

Cyclin D1, a pivotal oncoprotein, is overexpressed in approximately 65% of breast cancer cases [28], and is essential for cancer cell proliferation and tamoxifen resistance. Recent studies have shown that *Cyclin D1* performs distinct functions besides cell cycle regulation, including the alteration of gene expression within the local chromatin environment, cellular migration increase, and mitochondrial metabolism inhibition [29,30]. Dysregulated expression of *Cyclin D1* has been connected to hormonal therapy resistance development in breast cancer treatment [31].

Cyclin D1 interacts with a newly emerged long non-coding RNA *DILA1*, which is greatly upregulated in tamoxifen-resistant breast cancer cells. *DILA1* disrupts the phosphorylation of Cyclin D1 at Thr286 by directly binding to this site, subsequently blocking its degradation and leading to an increase in Cyclin D1 protein levels in breast cancer cells. A decrease in *DILA1* expression leads to *Cyclin D1* expression decline, which ends up in cancer cell growth suppression and restoration of sensitivity to tamoxifen, both in experimental cell cultures and animal models.

An increase in *DILA1* expression correlates with *Cyclin D1* levels enhancement and poor prognosis in breast cancer patients undergoing tamoxifen treatment. This shows the overlooked role of post-translational dysregulation of Cyclin D1 in tamoxifen resistance emergence. The identification of *DILA1* as a regulator of Cyclin D1 protein stability presents it as a promising therapeutic target for reducing *Cyclin D1* levels and combatting tamoxifen resistance in breast cancer therapy, in line with the findings of Shi et al. (2020), we also observed an increased expression of *Cyclin D1* and *DILA1* in MCF7-TR cells compared to MCF10A and MCF7-S cells [5].

The complicated interplay between the *Cyclin D1* and *PI3K/AKT/mTOR* signaling pathway has been involved in developing drug resistance. This interaction can be observed by targeting the *Cyclin D1/CDK4/6* complex. However, this therapeutic approach may initially prove effective; certain tumors are capable of activating the *PI3K/AKT/mTOR* pathway as a compensatory mechanism, thereby contributing to the emergence of drug resistance. On the contrary, pharmacological inhibition of the *PI3K/AKT/mTOR* pathway may result in the upregulation of *Cyclin D1* or other cell cycle regulatory proteins, effectively enabling cancer cells to circumvent the intended therapeutic effects [32,33]. Thus, a comprehensive understanding of these interconnected pathways is crucial in designing effective combination therapies to overcome drug resistance and improve clinical outcomes.

One of the key factors in drug resistance development is protracted exposure to tamoxifen, which is linked to the unusual activation of the *PI3K/AKT/mTOR* signaling pathways. The *PI3K/AKT/mTOR* pathway plays a crucial role in regulating cell proliferation and survival, especially in cancer cells. Inhibition of this pathway at different levels can have various

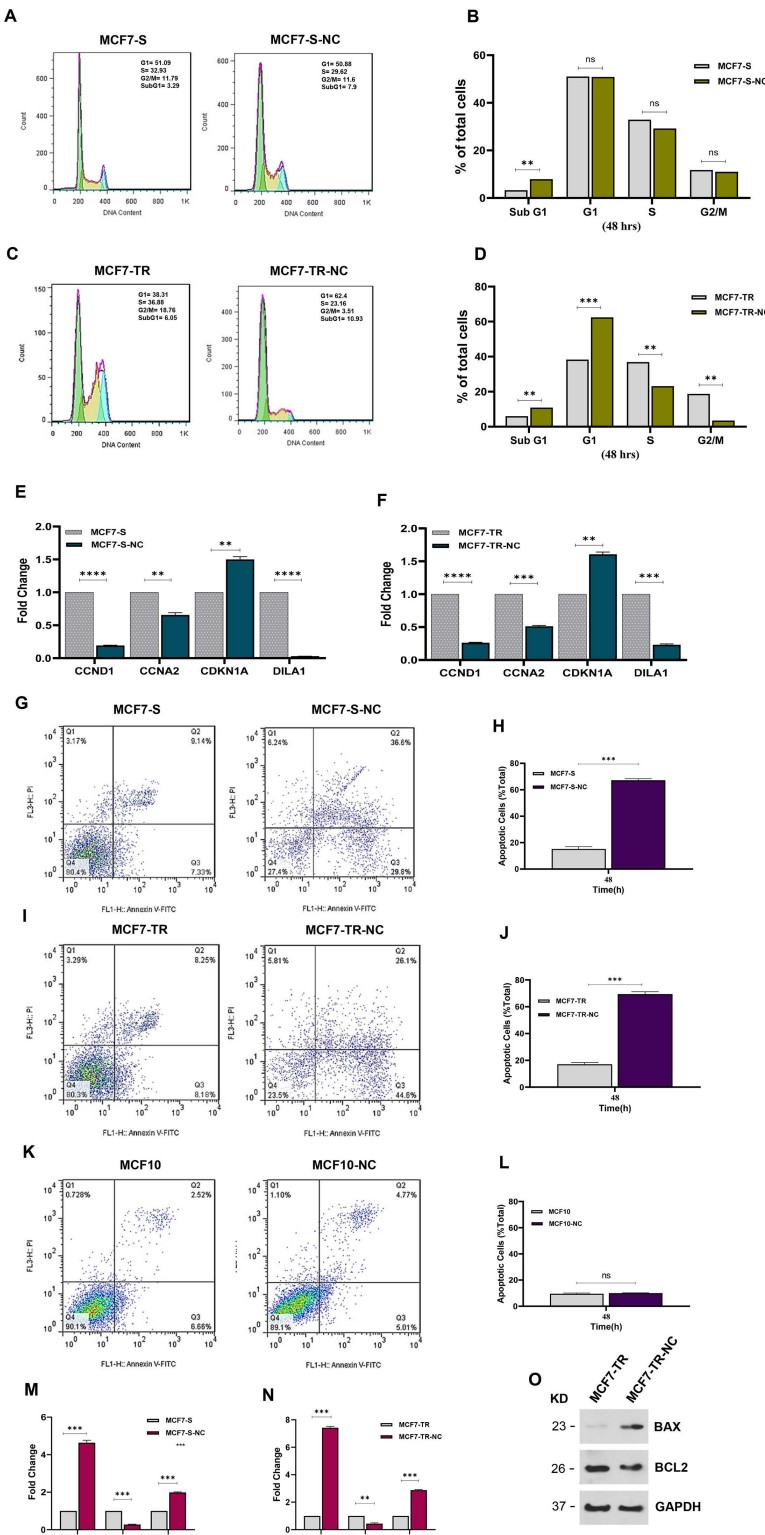

**Fig 5. Effects of nano-curcumin on cell cycle and apoptosis in breast cancer cells.** (A) Flow cytometry analysis with PI staining shows the effect of nano-curcumin on MCF7-S cells after 48h of treatment. (B) Displays the percentage of the cells in different phases of the cell cycle, following treatment with nano-curcumin. Sub-G1 cell population has increased due to nano-curcumin treatment after 48h. (C) PI staining flow cytometry analysis

of MCF7-TR cells treated with nano-curcumin in 48h. (D) The sub-G1 phase has risen in MCF7-TR-NC compared to MCF-TR cells. (E, F) Bar graphs show the relative expression levels of *CCNA2*, *CCND1,* and *CDKN1A* in breast cancer cell lines. The expression levels of *CCNA2* and *CCND1* were significantly decreased in both MCF7-S and MCF7-TR cells following nano-curcumin treatment. (G, I) Cell apoptosis assays reveal the apoptosis rate in MCF7-S and MCF7-TR cells after nano-curcumin treatment. (H, J) Apoptotic cell percentage was increased at 48h in the MCF7-S and MCF7-TR cell lines after being treated with nano-curcumin. (K, L) Cell apoptosis analysis revealed that the cytotoxic effect in MCF10A cells following 48 hours of treatment with nano-curcumin remained negligible compared to MCF10A- untreated. (M, N) The mRNA Expression levels of *BAX*, *BCL2* and P53 in MCF7-S and MCF7-TR cells after treated with nano-curcumin. *BAX* and P53 expression level significantly increase, whereas *BCL2* expression markedly decreases in cells treated with nano curcumin compared to untreated cells. (O) Western blot analysis confirms the upregulation of BAX and downregulation of BCL2 after nano-curcumin treatment. Data represent the mean of three independent experiments and are presented as means ± SEM. Statistical analysis was performed using student t-test or one-way ANOVA. ns = not significant, $*p < 0.05$, $**p < 0.01$, $***p < 0.001$, $****p < 0.0001$.

effects on cell behavior. PI3K inhibitors are being studied as a promising approach to preventing tamoxifen resistance. There are several clinical trials exploring various drug candidates that target the *PI3K/AKT/mTOR* pathway, implicating potent anticancer properties against resistance forms of breast cancer. While blocking the pathway can initially decrease cell proliferation and survival by increasing ER+ activity, the complex nature of the signaling cascade often causes the activation of compensatory mechanisms that allow the cells to develop resistance to single inhibitors. This underscores the complexity of targeting this pathway for therapeutic interventions [16,34].

In the present study, our findings demonstrate that nano-curcumin administration reduced *Cyclin D1* and *DILA1* expression in the MCF7-S and MCF7-TR cell lines when compared to untreated cells. The impact of nano-curcumin on the expression level of these genes led to the suppression of proliferation and an increase in apoptosis within the MCF7-S and MCF7-TR cells. We observed that nano-curcumin can induce cell death in MCF7-S cells and also facilitate the restoration of sensitivity to tamoxifen in MCF7-resistant cells. These findings underscore the indispensable role of *DILA1* in fostering cell proliferation and tamoxifen resistance.

Nano-curcumin can arrest cells in the G2/M phase in tamoxifen-resistant cells and induce apoptosis, which is done by increasing P53 phosphorylation, followed by Caspase 3 activation [35]. Our previous research on the efficacy of nano-curcumin in overcoming tamoxifen resistance [36] has been further supported by our recent findings.This study indicates that nano-curcumin administration leads to reduction in cell proportion of S and G2/M phase in both MCF7-s and MCF7-TR cell lines, demonstrating a novel mechanism of action through the regulation of *DILA1* and *Cyclin D1* gene expression. This discovery sheds light on a promising way to address tamoxifen resistance in breast cancer treatment.

This study indicates that nano-curcumin administration leads to reduction in S and G2/M phase in both MCF7-s and MCF7-TR cell lines.

Alternative signaling cascades involving *PI3K* and the fibroblast growth factor receptor pathway may be responsible for the upregulation of Cyclin D1 after tamoxifen resistance. Therefore, finding an alternative strategy is imperative to prevent Cyclin D1 activity in cancer and overcome resistance. The *CDK1/cyclinD6/4* and *PI3K/AKT/mTOR* pathways have become a desirable target for the treatment of many cancers, with resistant breast cancer being one of them, as tumorigenesis is largely dependent on uncontrolled cell cycle progression [37]. The interplay between the *Cyclin D1* and *PI3K/AKT/mTOR* signaling pathways in breast cancer is of paramount importance in tumor progression and drug resistance development. Therefore, a therapeutic approach that synergistically targets both pathways (dual-targeting strategy) may yield significant improvements in treatment efficacy [38].

Based on previous studies, curcumin plays an anticancer role through its impact on multiple signaling pathways. It has been shown that curcumin can significantly inhibit the proliferation of breast cancer cells by suppressing the phosphorylation of PI3K/AKT/mTOR signaling pathway-associated proteins. Moreover, it has been reported to induce autophagy and enhance lysosomal function by inhibiting the same pathway. Other research has highlighted the inhibitory effects of curcumin on the viability and epithelial-mesenchymal transition (EMT) in MCF7-TR cells, indicating its potential as a therapeutic agent for breast cancer treatment [39,40]. Consistent with previous research, our results suggest that nano-curcumin reduced cell proliferation by inhibiting *Cyclin D1* and *PI3K/AKT/mTOR* genes.

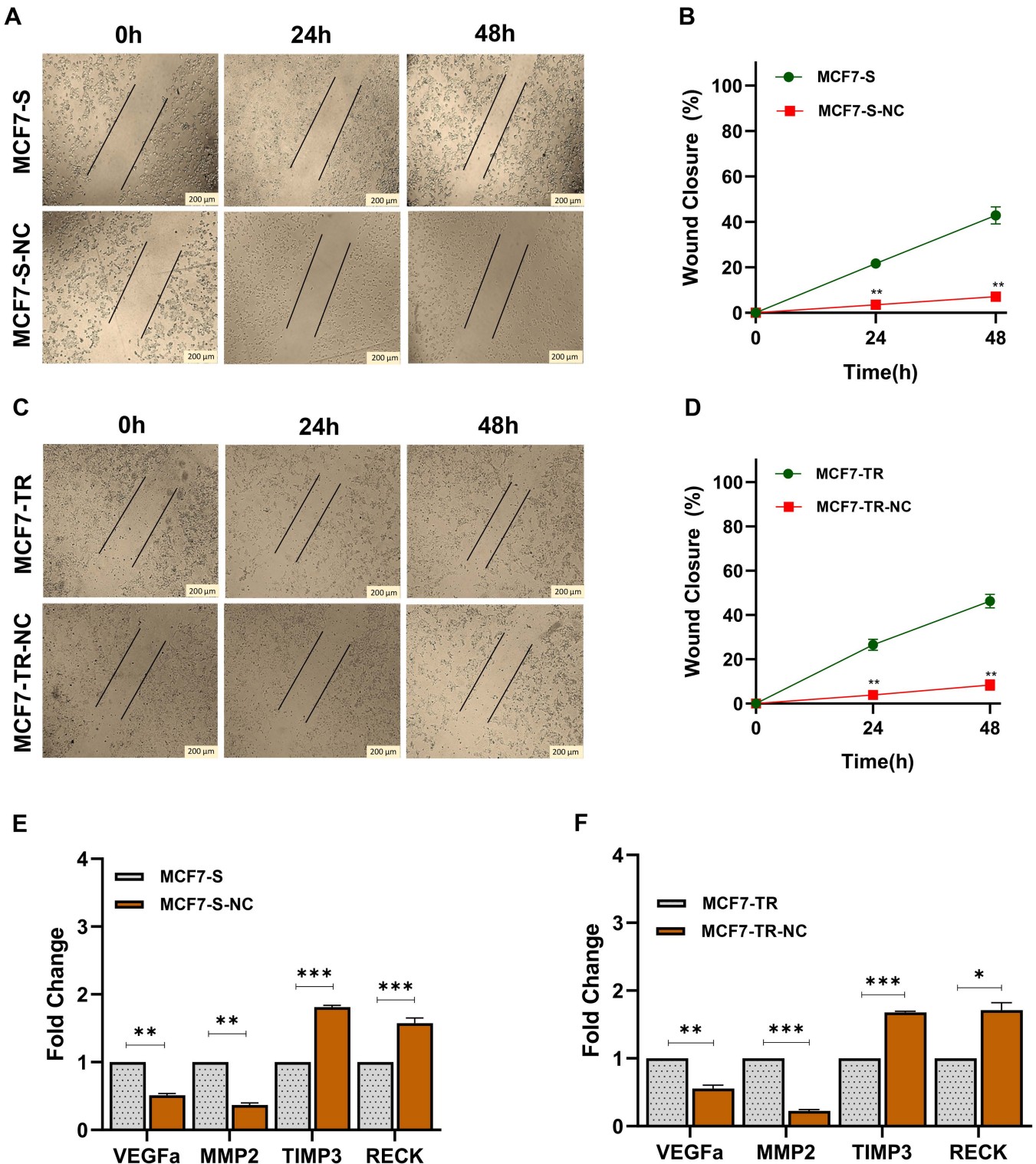

**Fig 6. Effects of nano-curcumin on cell migration.** (A, B) Wound healing analysis of MCF7-S and MCF7-S treated by nano-curcumin at 0h, 24h, and 48h post-scratching. Nano-curcumin in MCF7-S-NC cells significantly decreased cell migration in all time points tested. (C, D) Wound healing assay of MCF7-TR cells following treatment by nano-curcumin at 0h, 24h, and 48h. The nano-curcumin could decline the cell migration rate in MCF7-TR-NC cells. (E, F) *VEGFa, MMP2, TIMP3,* and *RECK* mRNA levels in MCF7-S and MCF7-TR breast cancer cells. The qRT-PCR results suggest that

nano-curcumin significantly decreases the migration potential of MCF7-S-NC and MCF7-TR-NC. Columns and points, mean of three different experiments. Statistical analysis was conducted using a student t-test or one-way ANOVA and means ± SEM were displayed. ns = not significant, *$p < 0.05$, **$p < 0.01$, ***$p < 0.001$, ****$p < 0.0001$.

These findings underlined the broad anticancer properties of curcumin for further investigation of its clinical potential in combination therapies targeting both the *Cyclin D1* and *PI3K/AKT/mTOR* pathways. In addition, treating MCF10A with nano-curcumin showed no significant cytotoxic effects, indicating that this nano formulation is well tolerated by non-cancerous cells. This observation further supports the specificity of nano-curcumin towards cancer cells while sparing their normal counterparts. Moreover, the expression levels of key proliferation-related genes, including *Cyclin D1, PI3K, AKT1, mTOR, and NF-κB* were examined in MCF10A cells, no significant differences were observed between treated and untreated groups. Collectively, these findings further highlight the cancer-specific activity of nano-curcumin.

By addressing the complex interplay between these pathways and their roles in drug resistance, curcumin may offer a promising strategy to improve treatment efficacy and overcome resistance mechanisms in breast cancer.

## Conclusion

In conclusion, this study indicates that nano-curcumin effectively reduces drug resistance while inhibiting cell proliferation, viability, and migration in both tamoxifen-sensitive and tamoxifen-resistant MCF-7 cell lines. Reduction of drug resistance via nano-curcumin treatment is achieved by decreasing the expression levels of *DILA1*, which subsequently diminishes the stability of the Cyclin D1 protein. Furthermore, nano-curcumin downregulates the expression of the *PI3K/AKT/mTOR* signaling pathway, thereby impacting cellular processes. Specifically, it suppresses the signaling pathways associated with *DILA1*, *Cyclin D1*, *PI3K/AKT/mTOR*, and *VEGF/MMPs* while promoting apoptosis and inducing cell cycle arrest in the sub-G1 phase in both sensitive and resistant MCF-7 cell lines.

The interplay between the *Cyclin D1* and *PI3K/AKT/mTOR* signaling pathways significantly contributes to drug resistance. Although targeting *Cyclin D1* has shown effectiveness, certain tumors may activate the *PI3K/AKT/mTOR* pathway as a compensatory mechanism, leading to resistance. The administration of nano-curcumin has been shown to reduce the expression of the *Cyclin D1* and *DILA1* genes, as well as downregulate the *PI3K/AKT/mTOR* signaling pathway, and furthermore, induce apoptosis in tamoxifen-resistant breast cancer cells (Fig 7). These findings suggest that the combination of nano-curcumin and tamoxifen may serve as a promising therapeutic strategy to overcome drug resistance in estrogen receptor-positive (ER+) breast malignancies.

We faced few limitations during this study. Primarily, the experiments were performed in vitro, and further in vivo investigations are necessary to confirm the observed effects in animal models. Ultimately, clinical trials will be required to evaluate the therapeutic potential of nano-curcumin in patients.

From a clinical perspective, our findings suggest that nano-curcumin may be considered as a potential adjuvant for ER+ breast cancer patients receiving tamoxifen therapy. Based on our results co-administration of nano-curcumin with tamoxifen can be a potential preventive approach for tamoxifen resistance, thereby it might lead to improvements in treatment efficacy.

Future research should address these aspects more comprehensively to strengthen the translational value of our findings.

## Supporting information

**S1 Fig. Original Western blot images.**
(PDF)

**S1 Table. The concentration of Tamoxifen.**
(PDF)

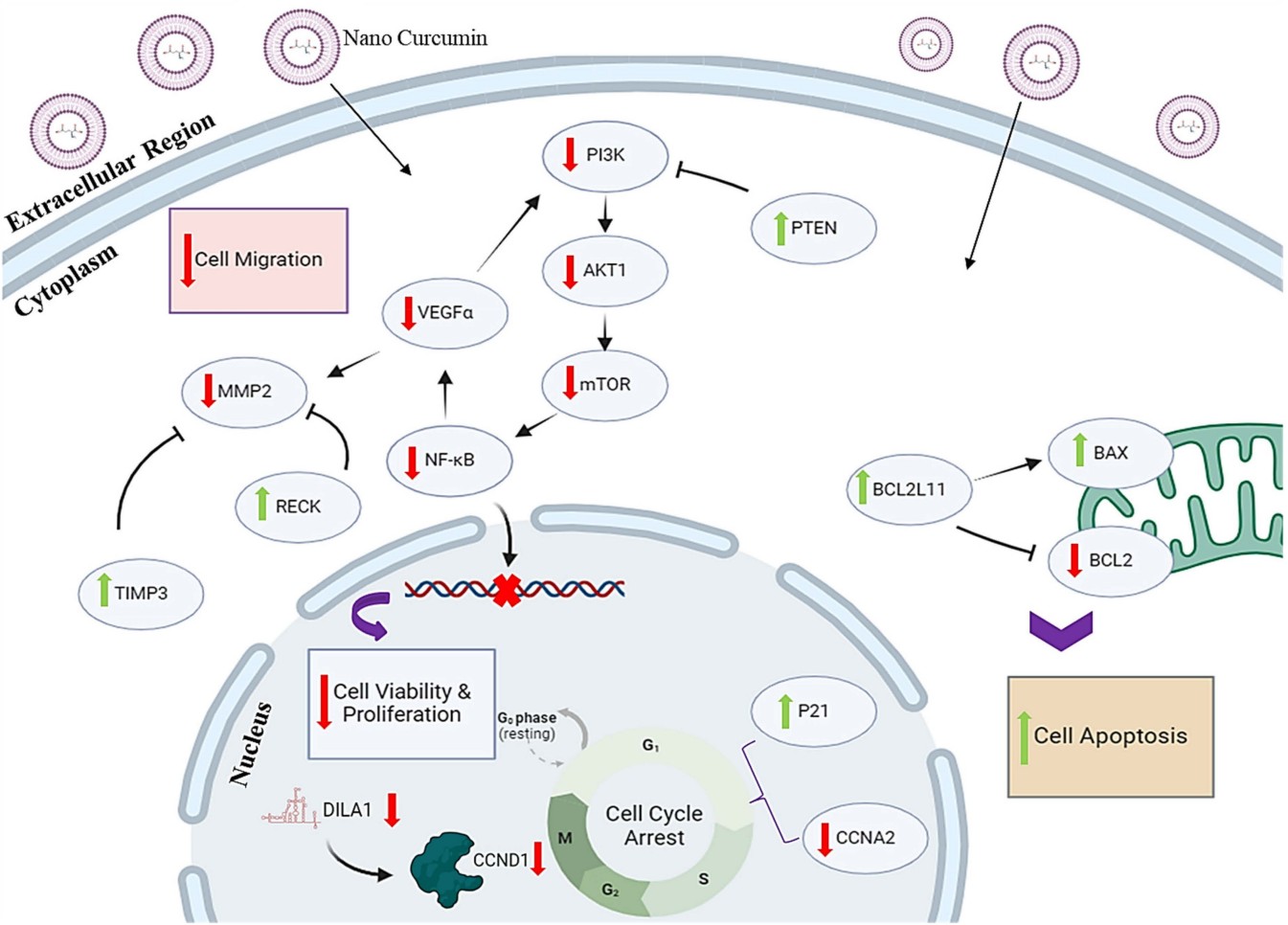

**Fig 7. Schematic representation of different pathways regulation by nano-curcumin.** The nano-curcumin, reduces the expression of Cyclin D1and DILA1 genes and PI3K/AKT/mTOR pathway activity which affects the biological process of cells, including cell proliferation and viability. Moreover, nano-curcumin regulates cell migration in this model by targeting αVEGF, RECK, TIMP3, and MMP2. nano-curcumin can reduce cell migration in MCF7-TR cells. As shown in this Figure, nano-curcumin can promote apoptosis by targeting BCL2L11, BAX, and also arrest the cell cycle by reduction of CCNA2 expression and up-regulation of CDKN1A (Parts of the figure are drawn by the BioRender site).

**S2 Table. The list of primer and oligo sequences.**
(PDF)

## Acknowledgments

The authors express gratitude to the members of the department of Genetics at Tarbiat Modares University.

## Author contributions

**Conceptualization:** Taraneh Givi, Majid Sadeghizadeh.

**Data curation:** Taraneh Givi.

**Formal analysis:** Taraneh Givi, Maryam Amirahmadi.

**Funding acquisition:** Taraneh Givi.

**Investigation:** Taraneh Givi, Majid Sadeghizadeh.

**Methodology:** Taraneh Givi, Fatemeh Mohajerani, Mohammadjavad Karimi Taheri, Majid Sadeghizadeh.

**Project administration:** Taraneh Givi, Fatemeh Mohajerani.

**Resources:** Taraneh Givi.

**Software:** Taraneh Givi.

**Supervision:** Taraneh Givi, Fatemeh Mohajerani, Sadegh Babashah, Majid Sadeghizadeh.

**Validation:** Taraneh Givi.

**Visualization:** Taraneh Givi.

**Writing – original draft:** Taraneh Givi, Fatemeh Mohajerani.

**Writing – review & editing:** Taraneh Givi, Fatemeh Mohajerani, Zahra Moazezi Tehrankhah, Majid Sadeghizadeh.

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
