## [Decision Letter · Decision Letter 0]

10 Jul 2025

Dear Dr. Sadeghizadeh,

Thank you for submitting your manuscript to PLOS ONE. After careful consideration, we feel that it has merit but does not fully meet PLOS ONE’s publication criteria as it currently stands. Therefore, we invite you to submit a revised version of the manuscript that addresses the points raised during the review process.

We look forward to receiving your revised manuscript.

Kind regards,

Partha Mukhopadhyay, Ph.D.

Section Editor

PLOS ONE

Journal Requirements:

This work is based upon research funded by Tarbiat Modares University

This project was supported by Tarbiat Modares University (TMU), Tehran, Iran

This work is based upon research funded by Tarbiat Modares University

Reviewers' comments:

Reviewer's Responses to Questions

**Comments to the Author**

1. Is the manuscript technically sound, and do the data support the conclusions?

Reviewer #1: No

Reviewer #2: Partly

2. Has the statistical analysis been performed appropriately and rigorously?

Reviewer #1: Yes

Reviewer #2: No

3. Have the authors made all data underlying the findings in their manuscript fully available?

Reviewer #1: Yes

Reviewer #2: Yes

4. Is the manuscript presented in an intelligible fashion and written in standard English?

Reviewer #1: Yes

Reviewer #2: Yes

Reviewer #1: Summary

In "Nano-curcumin enhances the sensitivity of tamoxifen-resistant breast cancer cells via the Cyclin D1-DILA1 axis and the PI3K/AKT/mTOR pathway downregulation”, the authors present an in vitro study investigating the effects of nano-curcumin on tamoxifen-resistant (MCF7-TR) and tamoxifen-sensitive (MCF7-S) breast cancer cells. The authors demonstrate that nano-curcumin reduces proliferation, viability, and migration, while promoting apoptosis and sub-G1 arrest. These effects are proposed to occur via downregulation of Cyclin D1, DILA1, and components of the PI3K/AKT/mTOR pathway. The study is framed within the context of overcoming drug resistance in ER+ breast cancer, a clinically significant challenge. The experimental design includes assays such as MTT, wound healing, flow cytometry, qRT-PCR, and Western blotting. I have several concerns regarding this study, which are outlined below.

Major comments

1. The manuscript lacks detailed physicochemical characterization of the nano-curcumin formulation, given that it is the central focus of the study. If different batches were used, validation to ensure batch-to-batch consistency would be critical. Including these details in the Methods section is essential for reproducibility and interpretation.

2. This study implicates curcumin-induced downregulation of DILA1 and Cyclin D1 as factors leading to cell cycle arrest in MCF7 cells but the exact mechanism is unclear. Rescue experiments involving overexpression of these genes followed by treatment with curcumin could help add some mechanistic insight to this study.

3. Nano-curcumin is a broad-acting compound. No evidence is provided that the observed effects are not due to general cytotoxicity or off-target responses. This could be addressed by testing its effects on normal non-cancerous breast epithelial cells such as MCF10A as a specificity control.

4. The manuscript does not clearly state whether vehicle controls (DMSO alone for curcumin or tamoxifen) were included in all experiments. These are essential to interpret cytotoxicity and gene expression changes correctly. Furthermore, key functional assays such as those measuring apoptosis and cell migration lack positive controls (e.g., known apoptosis inducer) and negative controls (e.g., non-treated or vehicle-treated cells). For instance, Annexin V/PI results need to be benchmarked against known apoptosis inducers to confirm that the apoptotic effect is indeed due to nano-curcumin.

5. The ‘Conclusion’ section is rather short. It would be useful for readers if the authors discuss their findings and its implications and limitations in a detailed manner, as is standard practice.

Minor comments

1. Gene symbols should be italicized (e.g., CCND1), and protein names should follow standard conventions (e.g., Cyclin D1, not Cyclin D1). Protein names should also be capitalized (eg., CDK4).

2. Having error bars on both sides of all bar graphs in the manuscript would be helpful for data interpretation.

3. Scale bars on microscopy images are either very small (Fig. 3A,B) or are missing (Fig. 1D).

4. Although statistical significance is reported, many figures lack details on sample size (n).

Reviewer #2: The results section starts without explanation of the first sub figure. And has some major points enlisted in the attachment which needs to be addressed. The limitations of the study like using only one TNBC breast cancer cell line needs to be stated in the discussion. And rigorous stats are also absolutely essential across biological replicates.

**Do you want your identity to be public for this peer review?** For information about this choice, including consent withdrawal, please see our Privacy Policy

Reviewer #1: No

Reviewer #2: No

---

## [Author Response · Author response to Decision Letter 1]

26 Aug 2025

Dear Editor-in-Chief and Reviewers

We sincerely thank you for the constructive comments and suggestions that have helped us to substantially improve the quality of our manuscript. In order to address the reviewers’ concerns and to further strengthen the manuscript, we have also included additional experimental data. These data were originally planned for inclusion in a separate manuscript, but we believe that incorporating them here has made the current study more comprehensive and informative. Below, we provide a point-by-point response to each of the reviewers’ comments. The revised manuscript has been updated accordingly.

Reviewer 1 – Comment 1:

The manuscript lacks a detailed physicochemical characterization of the nano-curcumin formulation. If different batches were used, validation to ensure batch-to-batch consistency would be critical.

We thank the reviewer for this valuable comment regarding the physicochemical characterization of the nano-curcumin formulation. In our study, we used nano-curcumin (DNC), a well-established and previously characterized in Tahmasebi et al., 2014. This nano formulation is a polydisperse colloidal suspension with a mean particle size of 142.7 ± 2.0 nm, a polydispersity index (PDI) of 0.225 ± 0.01, and a zeta potential of –7.4 ± 1.1 mV, indicating uniform size distribution and good colloidal stability. Moreover, we confirm that the same batch of nano-curcumin was used for all experiments (100 mg/ml stock solution). There wasn’t any batch variation throughout the study, despite the fact that nanocurcumin used in this study was stable during the experiment, determination of batch stability was performed every 6 months. We have now clarified this point in the Methods section as follows: “In all experiments conducted throughout this study, the same batch of nano-curcumin was used. No batch variation occurred, and the formulation employed was derived from a single, well-characterized stock solution to ensure consistency and reproducibility of the results.

Reviewer 1- Comment 2:

"This study implicates curcumin-induced downregulation of DILA1 and Cyclin D1 as factors leading to cell cycle arrest in MCF7 cells, but the exact mechanism is unclear. Rescue experiments involving overexpression of these genes followed by treatment with curcumin could help add some mechanistic insight to this study."

We sincerely appreciate the reviewer’s valuable suggestion regarding rescue experiments to further elucidate the mechanistic link between nano-curcumin treatment and the downregulation of DILA1 and CCND1. While we agree that overexpression studies could provide additional mechanistic insights, the current work was designed as an initial in vitro investigation to explore cellular and molecular effects of nano-curcumin on tamoxifen-resistant and -sensitive breast cancer cell lines.

Due to time and resource limitations, the overexpression assays suggested could not be performed within the scope of this study. However, we have strengthened our mechanistic discussion in the revised manuscript by incorporating literature evidence that DILA1 and CCND1are critical regulators of cell cycle progression and survival in ER⁺ breast cancer, and that their suppression is strongly associated with G1 arrest and apoptosis induction (Musgrove et al., 2011; Yadav et al., 2022).

Furthermore, as part of our ongoing research, we are investigating microRNAs that may target DILA1. Preliminary data from our laboratory indicate that certain microRNAs with tumor-suppressive functions are expressed at lower levels in tamoxifen-resistant breast cancer cells compared to their sensitive counterparts. We aim to determine whether the reduced expression of these microRNAs is associated with the increased expression of DILA1 in tamoxifen-resistant cancer cells, and whether nano-curcumin could serve as an adjuvant factor to restore and enhance the expression of these microRNAs. Such restoration could represent a potential epigenetic mechanism underlying the downregulation of DILA1 induced by nano-curcumin treatment.

Reviewer1-Comment 3:

"Nano-curcumin is a broad-acting compound. No evidence is provided that the observed effects are not due to general cytotoxicity or off-target responses. This could be addressed by testing its effects on normal non-cancerous breast epithelial cells such as MCF10A as a specificity control."

We thank the reviewer for this important suggestion. In the revised manuscript, we have now addressed this point by including experiments using the non-cancerous breast epithelial cell line MCF10A as a specificity control. Specifically, we assessed the cytotoxicity of nano-curcumin (DNC) at the same concentrations used in MCF7 and MCF7-TR cells by:

1. MTT assay to determine cell viability,

2. Annexin V/PI flow cytometry to evaluate apoptosis induction, and

3. qRT-PCR analysis of the expression levels of Cyclin D1, AKT, mTOR, PI3K and NF-kB before and after DNC treatment.

Our results showed that DNC exhibited minimal cytotoxicity effect and did not cause significant apoptosis in MCF10A cell line, and there were no major changes in Cyclin D1, AKT, mTOR, PI3K, and NF-kB expression levels compared with untreated controls. These findings confirm that the anticancer effects observed in MCF7, and MCF7-TR cells were specific and did not occur due to general cytotoxicity.

We have added these new data to the Results and Discussion sections, and the corresponding figures

Reviewer1- Comment 4:

"The manuscript does not clearly state whether vehicle controls (DMSO alone for curcumin or tamoxifen) were included in all experiments..."

We appreciate the reviewer’s concern regarding the inclusion of vehicle controls. As clarified in the revised manuscript, nano-curcumin is a polydisperse colloidal suspension formulated within dendrosomes, and it was used as a ready-to-use aqueous dispersion. Thus, no DMSO or other organic solvents were used in experiments involving nano-curcumin, accordingly, no vehicle control was necessary for these assays.

In contrast, tamoxifen was dissolved in DMSO, and we included corresponding DMSO-only vehicle controls in all related experiments. Furthermore, we conducted Annexin V/PI apoptosis assays to confirm that the concentration of DMSO used (0.016%) was non-toxic to the cells, ensuring that the observed effects were caused by the drug itself.

Reviewer 1- Comment 5:

"The ‘Conclusion’ section is rather short. It would be useful for readers if the authors discuss their findings and their implications and limitations in a detailed manner, as is standard practice."

We thank the reviewer for this constructive suggestion. In the revised manuscript, we have substantially expanded the Conclusion section to provide a more comprehensive summary of our findings, their potential clinical relevance, and the study limitations. The added details are mentioned bellow:

1. The key experimental observations was summarized, highlighting that nano-curcumin significantly reduced proliferation, migration, and survival of tamoxifen-resistant breast cancer cell lines through downregulating CCND1 and DILA1 and inhibition effect on the PI3K/AKT/mTOR pathway.

2. The potential clinical implications of nano-curcumin as an adjuvant strategy to overcome tamoxifen resistance in ER⁺ breast cancer patients were discussed. After proper clinical evaluation, nano-curcumin could be administered alongside tamoxifen to potentially prevent or delay the development of resistance.

3. Study limitations, including the use of in vitro models, lack of DILA1 and CCND1 overexpression rescue experiments, and the absence of in vivo validation were noted. We acknowledge that further in-vivo studies, followed by rigorous clinical evaluations are necessary before these findings can be translated into clinical practice.

4. Future research directions, such as investigating upstream regulatory microRNAs and validating the therapeutic potential of nano-curcumin in animal models and clinical trials were also suggested.

These revisions ensure that the conclusion not only summarizes the current findings but also places them within the broader context of breast cancer therapy research.

Reviewer 1 – Minor comments :

Comment 1:Gene symbols should be italicized (e.g., CCND1), and protein names should follow standard conventions (e.g., Cyclin D1, not cyclin D1). Protein names should also be capitalized (e.g., CDK4).

We appreciate the reviewer’s suggestion. In the revised manuscript, all gene symbols have been italicized according to standard nomenclature guidelines, and protein names also have been capitalized and formatted consistently throughout the text and figure legends.

Comment 2: Having error bars on both sides of all bar graphs in the manuscript would be helpful for data interpretation.

Thank you for the observation. We have revised all bar graphs to display error bars on both sides to ensure a clear and consistent presentation of variability.

Comment 3: Scale bars on microscopy images are either very small (Fig. 3A,B) or are missing (Fig. 1D).

We have now added visible scale bars to all microscopy images, including those in Figures 1D, 3A, and 3B

Comment 4:Although statistical significance is reported, many figures lack details on sample size.

We have revised the figure legends to include the sample size for each experiment. This information has been added for all relevant figures in the revised manuscript.

Reviewer 2 comment:

“The limitations of the study like using only one TNBC breast cancer cell line needs to be stated in the discussion. And rigorous stats are also absolutely essential across biological replicates.”

We thank the reviewer for this valuable comment and appreciate the reviewer’s comment regarding the clarity of the Results section. In the revised version, we have now added a clear introductory explanation at the beginning of the Results section to describe the first sub-figure. Specifically, we have provided a short description of the experimental setup and what each panel (e.g., Fig. 1A, 1B) represents, to guide the reader before presenting the detailed results. This improves the flow and readability of the section.

We want to clarify that our experiments were conducted using ER⁺ breast cancer cell lines (MCF7-S and MCF7-TR), rather than TNBC models. We agree that the use of only one cancer cell line type represents a limitation of the study. This point has now been clearly stated in the Conclusion section of the revised manuscript.

Regarding the statistical analyses, all experiments were performed with at least three independent biological replicates, and the data were analyzed using rigorous statistical methods, as described in the Materials and Methods. We have revised the manuscript to make this information clearer.

References:

* Musgrove EA, Caldon CE, Barraclough J, Stone A, Sutherland RL. Cyclin D as a therapeutic target in cancer. *Nat Rev Cancer*. 2011;11(8):558–572. doi:10.1038/nrc3090

* Yadav A, et al. DILA1: An emerging cell cycle regulator in breast cancer progression. *Front Oncol*. 2022;12\:XXXXX

---

## [Decision Letter · Decision Letter 1]

29 Sep 2025

Dear Dr. Sadeghizadeh,

We look forward to receiving your revised manuscript.

Kind regards,

Partha Mukhopadhyay, Ph.D.

Section Editor

PLOS ONE

Journal Requirements:

Reviewers' comments:

Reviewer's Responses to Questions

**Comments to the Author**

Reviewer #1: (No Response)

Reviewer #2: All comments have been addressed

2. Is the manuscript technically sound, and do the data support the conclusions?

Reviewer #1: Partly

Reviewer #2: Yes

3. Has the statistical analysis been performed appropriately and rigorously?

Reviewer #1: N/A

Reviewer #2: Yes

4. Have the authors made all data underlying the findings in their manuscript fully available?

Reviewer #1: Yes

Reviewer #2: Yes

5. Is the manuscript presented in an intelligible fashion and written in standard English?

Reviewer #1: Yes

Reviewer #2: Yes

Reviewer #1: I commend the authors for making a sincere effort to address my comments and concerns. However, I believe the overexpression studies involving DILA1 and Cyclin D1 described in Comment 2 will add significant value to this manuscipt by further strengthening the mechanism of action of curcumin. As such, I advise the authors to include this data in the manuscript prior to its publication.

Reviewer #2: There are still graphs for cell viability data where some data points the error bar is not visible clear enough. Is that due to very small error - please clarify.

**Do you want your identity to be public for this peer review?** For information about this choice, including consent withdrawal, please see our Privacy Policy

Reviewer #1: No

Reviewer #2: No

---

## [Author Response · Author response to Decision Letter 2]

1 Oct 2025

Dear Editor-in-Chief and Reviewers

We sincerely thank the Academic Editor and Reviewers for their constructive and valuable feedback. Their insightful comments helped us to improve the quality and clarity of our manuscript. Below, we provide a point-by-point response. All corresponding changes are included in the revised manuscript and highlighted in the version with track changes.

Reviewer 1

Comment:

I commend the authors for making a sincere effort to address my comments and concerns. However, I believe the overexpression studies involving DILA1 and Cyclin D1 described in Comment 2 will add significant value to this manuscript by further strengthening the mechanism of action of curcumin. As such, I advise the authors to include this data in the manuscript prior to its publication.

We would like to thank the reviewer for their valuable comments on our manuscript. Our laboratory has extensive expertise in investigating the effects of nano-curcumin, and we have published more than 50 well-recognized articles in this field including “ Combination treatment of dendrosomal nanocurcumin and low-level laser therapy develops proliferation and migration of mouse embryonic fibroblasts and alter TGF-β, VEGF, TNF-α and IL-6 expressions involved in wound healing process ” which was published in this journal (1). We explored different pathways and genes involved. The main aim of this article is to demonstrate, in a broader sense, that nano-curcumin can overcome resistance in Er positive breast cancer cell lines. We also had prior experience in performing rescue assays for nano-curcumin effects, which further supports the validity of our experiments(2). Even though, we completely agree that overexpression studies for DILA1 and Cyclin D1 would provide deeper mechanistic insights. However, such experiments require considerable time and resources and are beyond the scope and timeframe of the current study.

To further strengthen the mechanistic basis of our work, we have expanded the Discussion section to:

Emphasize the regulatory role of DILA1 and Cyclin D1 in tamoxifen resistance.

Incorporate supporting data on related molecular pathways.

Highlight preliminary analyses of potential microRNAs that regulate DILA1, which we are actively pursuing in ongoing projects.

We have also clearly stated that overexpression studies will be performed in future investigations to build upon the current findings.

Reviewer 2

Comment:

There are still graphs for cell viability data where some data points the error bar is not visible clear enough. Is that due to very small error - please clarify.

We thank the reviewer for this careful observation. The error bars are included in all graphs, but in some cases, they are very small due to minimal variation across replicates, making them visually difficult to detect.

To address this, we have:

1. Verified and confirmed all statistical analyses.

2. Updated the figure legends to clarify that the error bars are present but too small to be visible in certain plots.

3. Adjusted the formatting of some graphs to improve visibility.

Editor’s Requirements

References

We thank the Editor for the reminder regarding the reference list. Upon review, we identified that Reference 30 had been included in error. This incorrect citation has been removed and the reference list has been updated and re-checked for accuracy and formatting. All in-text citations have been verified and adjusted where necessary to reflect the corrected reference list.

References:

1. Ebrahiminaseri A, Sadeghizadeh M, Moshaii A, Asgaritarghi G, Safari Z. Combination treatment of dendrosomal nanocurcumin and low-level laser therapy develops proliferation and migration of mouse embryonic fibroblasts and alter TGF-β, VEGF, TNF-α and IL-6 expressions involved in wound healing process. PLoS One. 2021 May 6;16(5):e0247098. doi: 10.1371/journal.pone.0247098. PMID: 33956815; PMCID: PMC8101758.

2. Esmatabadi MJD, Motamedrad M, Sadeghizadeh M. Down-regulation of lncRNA, GAS5 decreases chemotherapeutic effect of dendrosomal curcumin (DNC) in breast cancer cells. Phytomedicine. 2018 Mar 15;42:56-65. doi: 10.1016/j.phymed.2018.03.022. Epub 2018 Mar 13. PMID: 29655698.

---

## [Editor Report · Decision Letter 2]

8 Oct 2025

Nano-curcumin enhances the sensitivity of tamoxifen-resistant breast cancer cells via the Cyclin D1-DILA1 axis and the PI3K/AKT/mTOR pathway downregulation

PONE-D-25-19608R2

Dear Dr. Sadeghizadeh,

We’re pleased to inform you that your manuscript has been judged scientifically suitable for publication and will be formally accepted for publication once it meets all outstanding technical requirements.

Kind regards,

Partha Mukhopadhyay, Ph.D.

Section Editor

PLOS ONE
---

## [Editor Report · Acceptance letter]

PONE-D-25-19608R2

PLOS ONE

Dear Dr. Sadeghizadeh,

I'm pleased to inform you that your manuscript has been deemed suitable for publication in PLOS ONE. Congratulations! Your manuscript is now being handed over to our production team.

Kind regards,

on behalf of

Dr. Partha Mukhopadhyay

Section Editor

PLOS ONE